# CONTRASTIVE VIDEO TEXTURES

## ABSTRACT

Existing methods for video generation struggle to generate more than a short sequence of frames. We introduce a non-parametric approach for infinite video generation based on learning to resample frames from an input video. Our work is inspired by Video Textures, a classic method relying on pixel similarity to stitch sequences of frames, which performs well for videos with a high degree of regularity but fails in less constrained settings. Our method learns a distance metric to compare frames in a manner that scales to more challenging dynamics and allows for conditioning on heterogeneous data, such as audio. We learn representations for video frames and probabilities of transitioning by fitting a video-specific bi-gram model trained using contrastive learning. To synthesize the texture, we represent the video as a graph where the nodes are frames and edges are transitions with probabilities predicted by our video-specific model. By randomly traversing edges with high transition probabilities, we generate diverse temporally smooth videos with novel sequences and transitions. The model naturally extends with no additional training to handle the task of Audio Conditioned Video Synthesis, when conditioned on an audio signal. Our model outperforms baselines on human perceptual scores, can handle a diverse range of input videos, and can combine semantic and audio-visual cues in order to synthesize videos that synchronize well with an audio signal.

## 1 INTRODUCTION

Generative Adversarial Networks (GANs) (Goodfellow et al., 2014) and Variational Autoencoders (VAEs) (Kingma & Welling, 2013) have achieved great success in generating images "from scratch". While one might have hoped that video generation would be a simple extension of image-generation methods, this has not been the case. A major reason is that videos are much higher dimensional than images, and producing correct transitions between frames is a difficult problem. While video generation (Vondrick et al., 2016; Mallya et al., 2020; Lee et al., 2019; Wang et al., 2018a;b) has shown some success, videos generated using such methods are relatively short and are unable to match the realism of actual videos. In comparison, classic non-parametric video synthesis methods from two decades ago, most notably Video Textures (Schödl et al., 2000), are much simpler and can often produce videos of arbitrary lengths.

In these models, a new plausible video is generated by stitching together snippets of an existing video. While video textures have been very successful on simple videos with a high degree of regularity, they use simple Euclidean pixel distance as a similarity metric between frames, which causes them to fail for less constrained videos containing irregularities and chaotic movements, such as dance or playing a musical instrument. They are also sensitive to subtle changes in brightness and often produce jarring transitions.

In this work, we propose Contrastive Video Textures, a non-parametric learning-based approach for video texture synthesis that overcomes the limitations of classic video textures. As in Schödl et al. (2000), we synthesize textures by resampling frames from the input video. However, as opposed to using pixel similarity, we *learn* feature representations and a distance metric to compare frames by training a deep model on *a single*

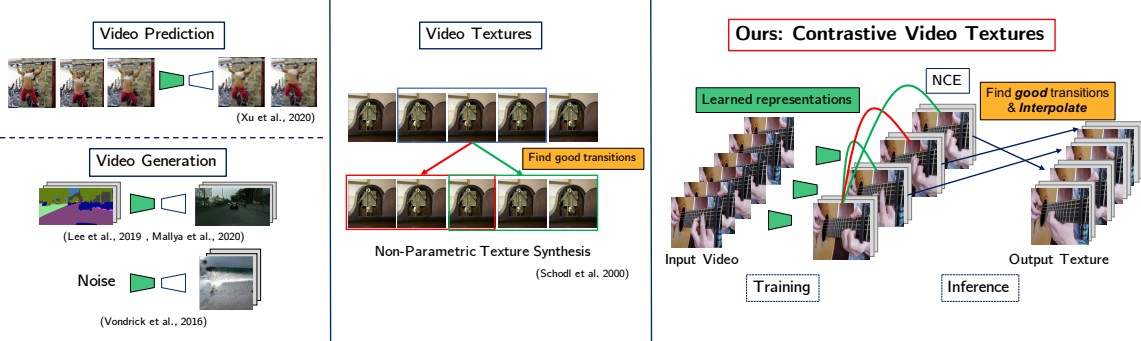

Figure 1: **Video Texture Synthesis.** Prior video prediction (Xu et al., 2020) and generation (Vondrick et al., 2016; Lee et al., 2019; Mallya et al., 2020) fail to generate long sequences and high resolution images. Classic video textures (Schödl et al., 2000) (middle) can generate infinite sequences by resampling frames, but uses fixed representations which are not robust to varying domains. Our method (right) learns a representation and non-parametric method for infinite video generation based on resampling frames from an input video.

*input video*. The network is trained using contrastive learning to fit an example-specific bi-gram model (*i.e.* a Markov chain).

To synthesize the video texture, we use the video-specific model to compute probabilities of transitioning between frames of the same video. We represent the video as a graph where the individual frames are nodes and the edges represent transition probabilities predicted by our video-specific model. We generate output videos (or textures) by randomly traversing edges with high transition probabilities. We additionally incorporate deep video interpolation into our contrastive video textures framework to suppress visual discontinuities and to allow for large transitions. Our proposed method is able to synthesize realistic, smooth, and diverse output textures on a variety of domains, including dance and music videos as shown at this website. Fig. 1 illustrates the distinction between video generation/prediction, video textures and our contrastive model.

We also extend our model to an audio conditioned video synthesis task. Given a source video with associated audio and a new *conditioning* audio not in the source, we synthesize a new video that approximately matches the conditioning audio. A demonstration of this task is shown at this link. We modify the inference algorithm to include an additional constraint that the predicted frame's audio should match the conditioning audio. We trade off between temporal coherence (frames predicted by the constrastive video texture model) and audio similarity (frames predicted by the audio matching algorithm) to generate videos which align well with the conditioning audio and are also temporally smooth.

We assess the perceptual quality of the synthesized textures by conducting human perceptual evaluations comparing our method to a number of baselines. In the case of unconditional video texture synthesis, we compare to the classic video texture algorithm (Schödl et al., 2000) and variations to this which we describe in Sec. 4. For the audio conditioning setting, we compare to three different baselines: classic video textures with audio conditioning, visual rhythm and beat (Davis & Agrawala, 2018), and a random baseline. Our results confirm that our method is perceptually better than all the baselines.

## 2 CONTRASTIVE VIDEO TEXTURES

We propose a non-parametric learning-based approach for video texture synthesis. At a high-level, we fit an example-specific bi-gram model (i.e. a Markov chain) and use it to re-sample input frames, producing a diverse and temporally coherent video. In the following, we first define the bi-gram model, and then describe how to train and sample from it.

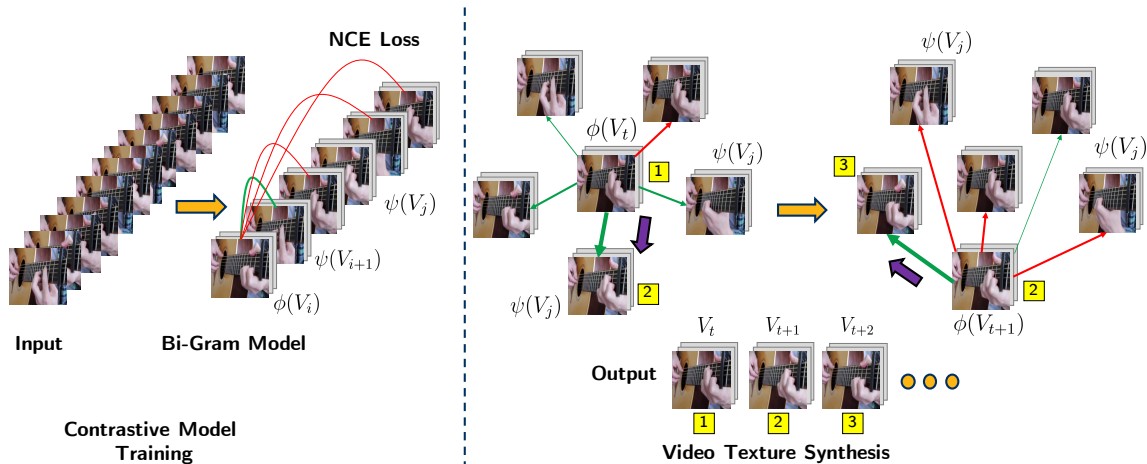

Figure 2: **Contrastive Video Textures:** We extract overlapping segments from the video and fit a bi-gram model trained using NCE loss which learns representations for query/target pairs such that given a query segment $V_i$, $\phi(V_i)$ is similar to positive segment $\psi(V_{i+1})$ and dissimilar to negative segment $\psi(V_j)$ where $j \in [1, ...N]$ and $j \neq i, i+1$.**Video Texture Synthesis.** During inference, we start with a random segment $V_t$ [1], compute $\phi(V_t)$ and $\psi(V_j)$ and calculate the edge weights as similarity between $\phi(V_t)$ and $\psi(V_j)$. We randomly traverse to (purple arrow) one of the edges that has high probability to reach [2]. We denote higher weight edges in green and lower weighted edges in red and the thickness correlates with the probability. No edge indicates zero similarity. The randomly chosen segment [2] is appended to the output as $V_{t+1}$ and the process is repeated (shown by the orange arrow) with [2] as the query.

Given an input video, we extract $N$ overlapping segments denoted by $V_i$ where $i \in [1, ...N]$, with a sliding window of length $W$ and stride $s$. Consider these segments to be the states of a Markov chain, where the probability of transition is computed by a deep similarity function parameterized by encoders $\phi$ and $\psi$:

$$P(V_{i+1}|V_i) \propto \exp(\text{sim}(\phi(V_i), \psi(V_{i+1}))/\tau) \tag{1}$$

Fitting the transition probabilities amounts to fitting the parameters of $\phi$ and $\psi$, which here will take form of a 3D convolutional network, by maximizing the log-likelihood of the sequence under the model:

$$\mathcal{L}(V, \phi) = \sum_{i=1}^{N} -\log P(V_{i+1}|V_i) = \sum_{i=1}^{N} -\log \frac{\exp(\text{sim}(\phi(V_i), \psi(V_{i+1}))/\tau)}{\sum_{j=1 [j \neq i, i+1]}^{N} \exp(\text{sim}(\phi(V_i), \psi(V_j))/\tau)} \tag{2}$$

where $\tau$ denotes a temperature term that modulates the sharpness of the softmax distribution. As the complexity increases with number of negatives in the denominator, for efficiency, we use negative sampling (Mikolov et al., 2013) to approximate in Eq 2. Fitting the encoder in this manner amounts to learning a video representation by contrastive learning, where the positive is the segment that follows, and negatives are sampled from the set of all other segments. The encoder thus learns features useful for predicting the dynamics of phenomena specific to the input video.

Given that we fit the model on a single video, it is important that we ensure there is enough entropy in the transition distribution in order to ensure diversity in samples synthesized during inference. While we assume that our input video sequence exhibits sufficient hierarchical, periodic structure to ensure repetition and multi-modality, we can also directly adjust the conditional entropy of the model through the softmax temperature term $\tau$. As we will see in Sec. 2, the encoder used for conditioning and prediction also plays a role in ensuring diversity in the sampling distribution. An overview of our method is provided in Fig. 2.

**Video Texture Synthesis.** To synthesize the texture, we represent the video as a graph, with nodes as segments and edges indicating the transition probabilities computed by our model. We randomly select a query segment $V_t$ among the segments of the video and set the output sequence to all the $W$ frames in $V_t$. Next, our contrastive model computes $\phi(V_t)$ and $\psi(V_j)$ for all segments in the video and updates the edges of the graph with the transition probabilities, given by $\text{sim}(\phi(V_t), \psi(V_j))$. The target segment with the highest transition probability is chosen as the positive segment. We then append the last $s$ number of frames in the positive segment to the output. This predicted positive segment $V_{t+1}$ is again fed into the network as the query and this is repeated to generate the whole output in an autoregressive fashion. This approach would regurgitate the original sequence, as the model was trained to predict $V_{i+1}$ as the positive segment given $V_i$ as the query. The edge with the maximum weight is always directed to the next segment in the video.

In order to introduce variance in the generated textures, we select segments which are similar to the positive segment $V_{i+1}$. First, we vary the temperature term $\tau$ to adjust the weights of the graph. The temperature term controls the entropy of the output distribution. A lower temperature would flatten the prediction probabilities/increase the entropy and reduce the difference in probabilities of the positive segment and segments similar to it. We then threshold the probabilities and set values to zero if they are less than $t\%$ of the max weighted edge connecting $V_t$ to any other node $V_j$, we set,

$$\text{sim}(\phi(V_t), \psi(V_j)) = 0 \,\forall\, j, \text{where}$$
$$\text{sim}(\phi(V_t), \psi(V_j)) < \max_{l=1,\dots,N} (\text{sim}(\phi(V_t), \psi(V_l))) - t\%$$

Next, we randomly select a frame to transition to from the edges with non-zero probabilities. This introduces variance in the generated textures and also ensures that the transitions are smooth and coherent.

**Video Encoding.** We use the SlowFast (Feichtenhofer et al., 2019) action recognition framework for encoding the video segments. We introduce two separate query and target multi-layer perceptrons to break the symmetry between the query and target embeddings. This ensures $\text{sim}(V_i, V_{i+1}) \neq \text{sim}(V_{i+1}, V_i)$ which allows us to learn the arrow of time.

**Interpolation.** For smoother transitions, we also conditionally interpolate between frames of the synthesized texture when there are transitions to different parts of the video. We use a pre-trained interpolation network of Jiang et al. (2018). We include results both with and without interpolation to show that interpolation helps with smoothing.

## 3 AUDIO-CONDITIONED CONTRASTIVE VIDEO SYNTHESIS

We extend Contrastive Video Textures to synthesize videos that match a conditioning audio signal. Given an input video and a conditioning audio $A^c$ we synthesize a new video that is synchronized with the audio. We extract $N$ overlapping segments from the conditioning audio, as before. We compute the similarity of the source audio segments $A^s$ to the conditioning audio segment $A^c$ by matching them in an embedding space and computing the similarity between the audio segments. We construct a transition probability matrix $T_a$ in the audio space as in Eq. 3.

$$T_a(i, j) = \text{sim}(\varphi(A_i^c), \varphi(A_j^s)) \tag{3}$$
$$T = \alpha T_v + (1 - \alpha) T_a \tag{4}$$

We compute the transition probabilities $T_v$ for the target video segments given the previous predicted segment using the contrastive video textures model (Eq. 2). The joint transition probabilities for a segment are formulated as a trade-off between the audio conditioning signal and the temporal coherence constraint as in Eq. 4.

Table 1: **Perceptual Studies** for Unconditional Video Textures and Audio Conditioned Video Synthesis.

(a) We show MTurk evaluators textures synthesized by all 5 methods and ask them to pick the most realistic one. We also report the chance evaluators chose *any* of the variation of the classic model.

| Method | Preference % |
|---|---|
| Classic | 3.33 ± 2.42 % |
| Classic Deep | 6.66 ± 3.37 % |
| Classic+ | 10.95 ± 4.22 % |
| Classic++ | 9.52 ± 3.97 % |
| Any Classic | 30.48 ± 6.22 |
| Contrastive | **69.52** ± 6.22 % |

(b) **Unconditional: Real vs. Fake study.** We show evaluators a pair of videos (generated and real video) without labels, ask them to pick the real one. Our method fools evaluators more times than Classic.

| Method | Real vs. Fake |
|---|---|
| Classic++ | 11.4 ± 4.30% |
| Classic+ | 15.7 ± 4.92 % |
| Contrastive | **45.7** ± 4.3% |

(c) **Conditional: Real vs. Fake study.** We show evaluators a pair of videos (generated and real video) without labels and ask them to pick the real one. Our method fooled evaluators more often than did baselines.

| Method | Real vs Fake |
|---|---|
| Random Clip | 15.33 ± 5.76% |
| Audio NN | 20.4 ± 6.63% |
| Contrastive | **26.74** ± 6.14% |

**Audio encoding.** We embed the audio segments using the VGGish model (Hershey et al., 2017) pretrained on AudioSet (Gemmeke et al., 2017). We remove the last fully connected layer from the model and use the output of the final convolutional layer as audio features. We describe details of the implementation of our method in Sec. A.1.

## 4 EXPERIMENTS

We curate a dataset of 70 videos from different domains such as dance and musical instruments including piano, guitar, suitar, drums, flute, ukelele, and harmonium. A subset of these videos were randomly sampled from the PianoYT dataset (Koepke et al., 2020) and the rest were downloaded from YouTube. We first worked with 40 of the 70 videos and used those to tune our hyperparameters. We then tested it on the remaining 30 without any tuning. Our dataset consists of both short videos which are 2-3 minutes long and long videos ranging from 30 - 60 mins[1]. We conduct perceptual evaluations on Amazon MTurk to qualitatively compare the results from our method to different baselines for both the aforementioned cases.

### 4.1 UNCONDITIONAL VIDEO TEXTURE SYNTHESIS

To show the effectiveness of our method, we compare our results to the classic video textures algorithm (Schödl et al., 2000). Additionally, we compare to three variations of the algorithm. Classic+ appends multiple frames to the output sequence instead of a single frame, Classic++ adds a stride while filtering the distance matrix and Classic Deep uses ImageNet pretrained ResNet features instead of raw pixel values. The algorithm and its variants are described in Sec. A.2.

Table 1a reports the results from a perceptual study on Amazon MTurk where evaluators where shown textures generated by all five methods and asked to choose the one they found most realistic. Our contrastive model surpasses all baselines by a large margin and was chosen 69.52% of the time. Since the classic models are similar, we also report all variations of classic combined. They chosen 30.48% of the time.

We include qualitative video results for Classic, Classic+, Classic++ and contrastive in Fig. 3. Video in Fig. 3a produced by our contrastive method is most realistic and has seamless transitions. We notice in Fig. 3b that

---

[1]We will release the videos along with the trained models for each, along with a reference implementation of our method.

the classic algorithm often transitions to frames in and around the target, producing textures which are jarring. Classic+ shown in Fig. 3c has slightly improved quality compared to classic but is still nowhere close to our contrastive method. Classic ++ shown in Fig. 3d shows no significant improvement.

Additionally, we conduct real vs. fake studies where the evaluators are shown the ground truth video and synthesized texture and asked to pick the one they think is real. Our method is able to fool evaluators 45.7% of the time whereas the best baseline (Classic+) is able to fool the evaluators only 15.7% of the time.

Both the qualitative and quantitative comparisons clearly highlight the issues with the classic model and emphasize the need to learn the feature representations and the distance metric as we do in our contrastive method.

|  |  |  |  |
|:--:|:--:|:--:|:--:|
| (a) Contrastive | (b) Classic | (c) Classic+ | (d) Classic++ |

Figure 3: To play the videos, please view in Acrobat Reader and click on the figure. Videos are also included at this [website]. Qualitative comparison of our method (Contrastive) to the baselines Classic, Classic+ and Classic++ are shown. The red bar at the bottom indicates the part of the original video being played. The results from all 3 baselines are choppy. Our method selects good transitions in the video as can be seen by the red bar moving but the transition is seamless.

### 4.2 AUDIO CONDITIONED CONTRASTIVE VIDEO TEXTURES

For audio conditioned video synthesis, we choose a subset of 30 videos from the above 50 and randomly paired them up with songs from the same domain (*e.g.* a source piano video is paired with a conditioning audio of a piano). Using this strategy, we created 50 source video - conditioning audio pairs. As described in Sec. 3, we extend the contrastive method to synthesize textures given a conditioning audio signal. We compare audio conditioned video textures synthesized by our method to four baselines and report results from a perceptual evaluation.

**Random Clip.** To show it is easy to identify when a video and the corresponding audio are out of sync, we randomly choose a portion of the source video to match the conditioning audio.

**Video Textures with Audio Conditioning.** We add audio conditioning to the classic video textures algorithm. For this, we divide the conditioning audio into segments and find nearest-neighbours in the source audio. Then we combine these distances with the distance matrix $D$ calculated by video textures.

**Visual Rhythm and Beat.** We use the approach of Davis & Agrawala (2018) to synchronize the source video with the audio beats. This method works by changing the pacing of the video (slowing it down and speeding it up) so that the visual and audio beats are more closely aligned.

**Audio Nearest Neighbours.** We include comparisons to a nearest neighbor baseline that works by computing the similarity between the conditioning audio signal and segments of the source audio of the same length, and then choosing the video clip of the closest match.

We conduct perceptual studies comparing the audio conditioned video textures synthesized by our contrastive model and all of the baselines. The evaluators were shown two videos with the same conditioning audio, one synthesized by our method and the other by the baseline. They were asked to pick the video that they felt was more in sync with the audio. Our method was chosen 92% of the time when compared with Classic+Audio,

84% of the time when compared with VRB, 70% of the time when compared with Random Replay and 66% of the time when compared with Audio NN. As shown in Tab. 1c, we conducted a real vs. fake study comparing the ground truth videos with the synthesized videos from contrastive and the two best baselines (Random Clip and Audio NN). While Random Clip and Audio NN beat the ground truth only 15.33% and 20.4% respectively, our method was able to fool evaluators 26.74% of the time.

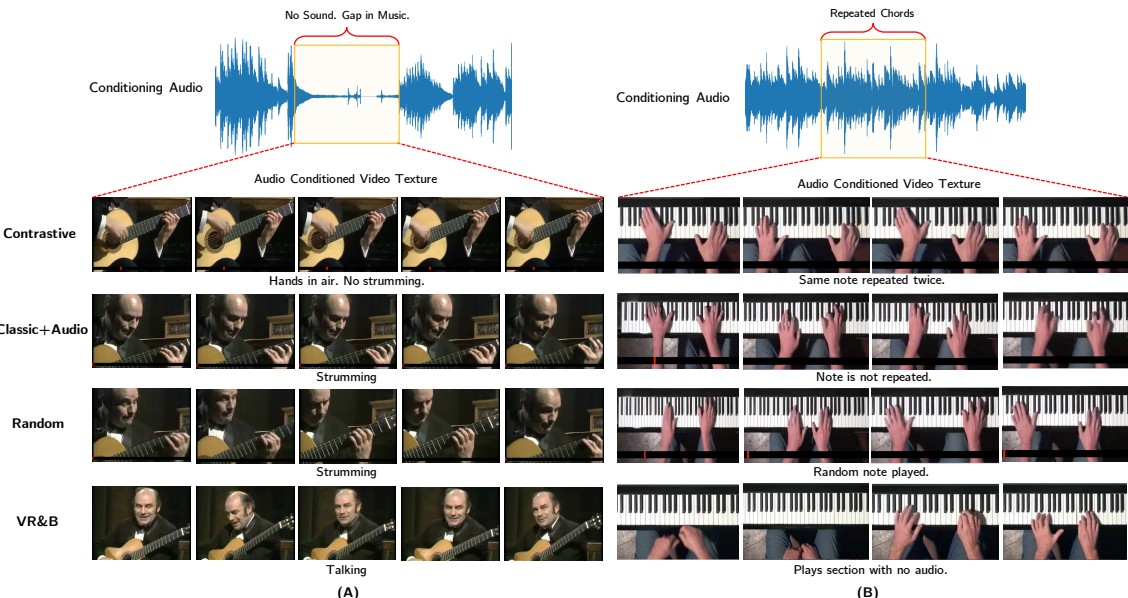

Figure 4: Qualitative comparison of audio conditioned video textures synthesized by Classic, Random, Visual Rhythm and Beat (VRB) and our Contrastive model. **(A)** The conditioning audio waveform shows a gap in the audio where no music is being played. Our model is able to pick up on that and the corresponding video that is synthesized has hands in the air and no strumming. However both Random and Classic+Audio show strumming, which is not in sync with the audio. The result using VRB shows the person talking. **(B)** The conditioning audio waveform has the same chord repeated twice. The video synthesized by our model reflects this, and we observe the same frames (1 and 2) repeated again. Classic+Audio plays the note just once and Random plays a different note. VRB result contains a region without audio where the person isn't playing anything, which is out of sync with the conditioning audio.

We show qualitative results comparing our method to the five baselines described above in Fig. 4 and also include additional results at this [website]. In Fig. 4 (A), the conditioning audio signal has a gap/break in the audio where no music is being played. We see the output produced by the contrastive model is semantically meaningful and aligns best with the audio. Random Clip chooses a random segment which has strumming and thus fails to align with the audio. Similarly class+audio chooses frames that don't correlate with the audio. VRB doesn't capture semantics as it only speeds up or slows down the video to better match the audio beats. Similarly, in Fig. 4(B) we see that our contrastive method is able to pick up on repeated chords in the conditioning audio signal while no other method is able to do that. Furthermore, we also illustrate the importance of adding video interpolation through examples show here. Through more examples listed at this website we show that the videos synthesized by contrastive model are more in sync with the conditioning audio. For example, it identifies gaps in the audio, repeated chords, and change of pace. Our model is better at learning audio-visual correlations.

## 5 RELATED WORK

**Texture Synthesis.** All texture synthesis methods aim to produce textures which are sufficiently different from the source yet appear to be produced by the same underlying stochastic process. Texture synthesis methods can be broadly classified into two categories: non-parametric and parametric. Non-parametric methods focus on modeling the conditional distribution of the input images and sample information directly from the input. The sampling could be done pixel-wise (Efros & Leung, 1999; Wei & Levoy, 2000) or patch-wise (Efros & Freeman, 2001; Kwatra et al., 2003) for image texture synthesis. Inspired by these works, (Schödl et al., 2000) proposed a non-paramteric approach for synthesizing a video texture with by finding novel, plausible transitions in an input video. (Schödl & Essa, 2001; 2002; Efros et al., 2003) were interesting extensions of the same. (Wei et al., 2009) provides an extensive review of example-based texture synthesis methods. Parametric approaches, on the other hand, focus on explicitly modeling the underlying texture synthesis process. (Heeger & Bergen, 1995; Portilla & Simoncelli, 2000) were the first to propose parametric image texture synthesis by matching statistics of image features between source and target images. This later inspired (Gatys et al., 2015), which used features learned using a convolutional neural network for image texture synthesis.

**Video Generation and Video Prediction.** The success of (GANs) (Goodfellow et al., 2014) and Variational Autoencoders (VAEs) (Kingma & Welling, 2013) in image generation (Zhu et al., 2017; Karras et al., 2019; Park et al., 2019) inspired several video generation methods, both unconditional (Vondrick et al., 2016; Clark et al., 2019; Tulyakov et al., 2018; Saito et al., 2017) and conditional (Wang et al., 2018b; Chen et al., 2019; Wang et al., 2019; Zhou et al., 2019; Gafni et al., 2020; Mallya et al., 2020). A common type of conditional video synthesis includes future frame prediction given past frames (Denton & Birodkar, 2017; Srivastava et al., 2015; Kalchbrenner et al., 2017). Even the most recent video prediction (Ye et al., 2019; Xu et al., 2020) techniques produce blurry outputs and fail to generate frames beyond a few seconds. These methods are far from generating realistic videos and oftentimes produce outputs which are blurry and low-resolution, especially in the unconditional case. This is because videos are higher dimensional and modeling spatio-temporal changes and transition dynamics is more complex. As such, these methods are expected to fail when applied to our task of video texture synthesis. There are also a few recent works which condition the video generation on an input signal such as text (Li et al., 2018), or speech (Oh et al., 2019; Kim et al., 2018; Ephrat et al., 2018), or a single image (Shaham et al., 2019). Our method is similar to SinGAN (Shaham et al., 2019) in that we train our network on a single input, though on a video instead of an image and without an adversarial loss.

**Contrastive Learning.** Recent contrastive learning approaches (Hénaff et al., 2019; Chen et al., 2019; 2020; He et al., 2020) have achieved success in classic vision tasks proving the usefulness of the learned representations. (Misra et al., 2016) train a network to determine the temporal ordering of frames in a video and (Wei et al., 2018)'s self-supervised model learns to tell if a video is playing forwards/backwards. Here, we use contrastive learning to fit a video-specific bi-gram model. Our network maximizes similarity between learned representations for the current and next frame. Unlike (Oord et al., 2018), our goal is not to generate frames from latent representations, but rather to resample from the input video.

## 6 CONCLUSION

We presented contrastive video textures, a learning-based approach for texture synthesis applied to audio conditioned video generation. Our method fits an input-specific bi-gram model to capture the dynamics of a video, and uses it to generate diverse and temporally coherent textures. Further, we introduce the task of audio conditioned video texture synthesis as a useful application of video textures. We show that our model outperforms a number of baselines on perceptual studies. We hope this work inspires research in texture synthesis based generation.

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

## A  APPENDIX

### A.1  IMPLEMENTATION DETAILS

We divide the video into overlapping segments using a window of length 0.5 seconds and a stride of 0.2 seconds. Depending on the frame rate of the video, this yields segments with varying number of frames.

**Video Encoding.** Each of these segments is then encoded by SlowFast model into $\mathbb{R}^{512}$. Next the query and target are passed through two separate MLPs, each consisting of 3 linear layers interspersed with ReLU activations. The MLP maintains the size of the embedding such that the final outputs, $\phi(S)$ and $\psi(S)$ are in $\mathbb{R}^{512}$. We initialize the SlowFast model with weights pretrained on Kinetics dataset and fine tune the whole network end-to-end. We use SGD optimizer with a learning rate of 1e-4 for the Slow Fast model and a rate of 1e-3 for the MLP.

**Audio Encoding.** The VGGish model is initialized with weights pretrained on AudioSet. The learned audio representations for the source audio segments $\varphi(A^c)$ and the conditioning audio segments $\varphi(A^c)$ are in $\mathbb{R}^{128}$.

**Interpolation.** We typically set the number of interpolated frames to be added to be 4. This increases the FPS of the synthesized video by a factor of 3 (i.e. 2 frames is converted to 4). When there is no jump, the frames are repeated 3 times, to ensure the overall FPS of the video is the same.

**Temperature tuning and threshold.** For training the contrastive video texture model, we experimented with multiple values of temperature ($\tau$) and found 0.1 to work the best. At test time, setting the temperature to 0.1 and threshold ($th$) to 0.0 regurgitates the original video. Increasing the temperature and threshold increases the entropy and allows for more random transitions in the output. We found that the number of transitions vs the temperature is fairly constant across all videos and include details in Sec. 4. We found a temperature of $0.3$ and a temperature of $th$ is optimal for synthesizing videos which are temporally smooth yet different from the original video.

**Combining $T_a$ and $T_v$.** Smaller values of $\alpha$ allow for better audio-video synchronization but at the cost of continuity in the video. For most results reported here, we set $\alpha$ to either 0.5 or 0.7.

### A.2  UNCONDITIONAL VIDEO TEXTURES: BASELINES

We first provide an overview of the classic video texture algorithm introduced in (Schödl et al., 2000) followed by the descriptions of the baselines.

**Classic Video Textures** The classic video textures algorithm proposed in (Schödl et al., 2000) computes a distance matrix $D$ of pairwise distances between all frames in the video. The distance is computed as the L2-norm of the difference in RGB values between pairs of frames. Next, the distance matrix $D$ is filtered with a 2 or 4-tap filter with binomial weights to produce matrix $D'$. The stride used while filtering is 1. If the input video is short, oftentimes this approach would not be able to find good transitions from the last frame and reaches a dead end. To avoid this, they use Q-learning to predict the anticipated (increased) "future cost" of choosing a given transition, given the future transitions that such a move might necessitate. This gives rise to $D''$. The transition probabilities $P''$ are computed from $D''$ as $P''_{i,j} = \exp(-D''_{i+1,j}/\sigma)$.

To synthesize a texture, a frame $i$ is chosen at random. This is added to the output sequence of frames. After displaying frame $i$, the next frame $j$ is selected according to $P_{i,j}$. To improve the quality of the textures and to suppress non-optimal transitions they adopt a two step pruning strategy. First, they choose the optimal transition with the maximum transition probability, next they set all probabilities below some threshold to zero and pick a random transition from the non-zero probabilities. The output sequence is generated one frame at a time.

We generate textures using the algorithm above. Following the convention in (Schödl et al., 2000), we set sigma to be a small multiple of the average (non-zero) values in the distance matrix. We tune this small multiple and the threshold on the train set and use the same values on the test set.

For an apples-to-apples comparison, we fix some of the shortcomings of the classic algorithm and compare to these modified versions described below.

**Classic+.** During inference, the number of frames appended to the output texture is the stride with which the initial video was segmented. While this stride is 1 for the classic algorithm, it is greater than 1 for our contrastive method. To ensure the difference in perceptual quality isn't due to just the changes in stride length, we modify the classic algorithm to increase the stride during inference to be the same as our contrastive model. The distance matrix is still computed pairwise between frames but instead of appending a single frame, we append stride number of frames to the output. This stride is set to be the same value as our contrastive model.

**Classic++.** To further reduce the gap between classic+ and contrastive method, we apply a stride ¿ 1 while filtering the distance matrix $D$ with the tap filter. This is equivalent to the approach we use in contrastive, which is dividing the video into overlapping segments of window $W$ and stride $s$.

**Deep Classic.** Additionally, we also tried replacing the frame-wise features in the classic algorithm with learned representations from a pre-trained resnet.

## A.3    TRANSITION PROBABILITIES

We compare the **transition probability matrices** generated by both classic and contrastive methods. Fig. 5a shows the transition probability matrices for two different videos generated by Classic (1a, 2a) and Contrastive (1b, 2b) methods. It can be observed from the diagonal lines in the figure that the classic method assigns the same value to multiple frames whereas our method picks up on subtle differences and assigns different scores. This emphasizes that the distance metric learned by our method is better at distinguishing frames.

Fig. 5b shows the variation in the number of transitions with sigma for the classic technique and with temperature for the contrastive technique. Number of transitions increases linearly with temperature for contrastive method whereas for the classic technique we found no such correlation. Moreover, a temperature of 0.3 and a threshold of 0.01 results in  15-20 jumps across all videos. There was no such strong correlation for the classic technique, making it necessary to tune hyperparameters on a per video basis.

**Failure Cases.** As shown here, our method fails when the changes in background/lighting are too large to be smoothened by a interpolation model. This is observed in dance videos where there's large movements across segments. VRB and Random Replay work well in such cases. VRB is designed to align dance moves with the music and hence clearly performs the best. Random works well too as good music and good dancing go well together.

Fig. 6 shows some transitions in the video textures generated by contrastive model.

Figure 5: **Left.** Transition probability matrix for two different videos (in each row) for both classic and contrastive methods. **Right.** Number of transitions vs Sigma for Classic and Number of transitions vs Temperature for Contrastive.

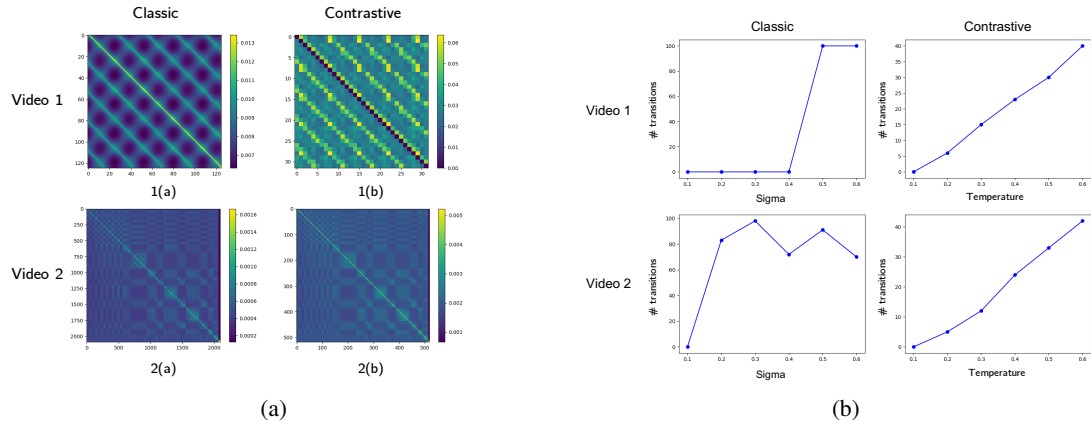

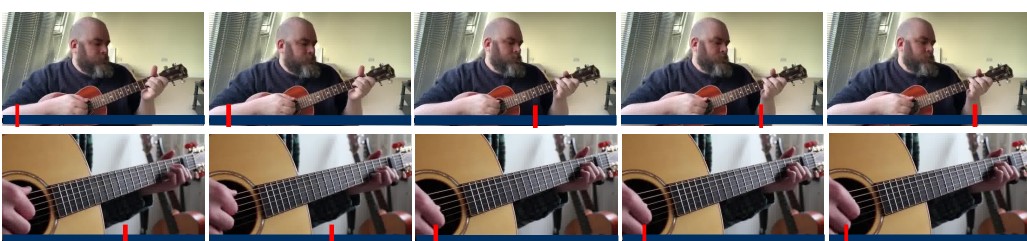

Figure 6: The figure shows frames from two different videos synthesized by our method. Red bar indicates position of the original video being played. The transition happens at the third frame and is seamless in both cases. The first is a forward jump and the second is a backward jump.

