# OpenReview forum: "Contrastive Video Textures"
_ICLR.cc/2021/Conference — Reject_

### Official Review · AnonReviewer4 · 2020-10-20
**Promising results, more discussions/comparisons are requested**

**Rating:** 6
**Confidence:** 3

**Review:**

In this paper, the authors proposed a non-parametric approach for video generation, i.e., video frame (un)conditional resampling. The proposed method is inspired by Video Textures (Sch¨odl et al., 2000), which synthesizes new videos by stitching together snippets of an existing video. Comparing to existing 'video textures' methods, the authors mainly made two improvements/contributions. (i) a new pipeline for modeling and calculating probabilities of transitioning between frames of the same videos. Specifically, giving a video clip, the authors first extract overlapping segments from it and fit a bi-gram model. The adjacent segments are regarded as positive pairs with high transitioning probability, yet other random sampled pairs are negative pairs. Similar to contrastive learning works, NCE loss is utilized to train the bi-gram model. (ii) Extending the model to a conditional situation and performing the task of audio conditioned video synthesis. The authors made a trade-off between the audio conditioning signal and the learned transition probabilities. Finally, experiments including multiple qualitative resampled videos and quantitative user studies were provided. And the proposed method demonstrated promising performance.

Overall, this work is well organized and easy to follow. The motivation is clear and ideas are simple and useful. According to the provided video examples, the proposed method achieved great performance. However, I still have some concerns:

In the experiments of unconditional setting, the authors included Classic, Classic+, and Classical++ as their comparisons. However, all recent (here, I mean in the past decade) methods that related to video generation are missed. First, although there may not exist any resampling method that can directly perform the video texture synthesis, I believe many related graph-based methods could be used to model the transition probabilities of frames. I think the authors may at least discuss these works. Second, although the authors pointed out that the video resampling (textures) strategy is different from the recent generation-based strategy, they should provide visual results/comparisons to support their claims. In recent years, some methods were proposed to synthesize videos with dynamic textures, such as flame, wave, flash, etc. I am interested in if the proposed method can perform these tasks. In addition, the authors may further discuss the limitations of the proposed method.

---

> ### Author Response · Authors · 2020-11-13
> **Thank you for the constructive feedback. We've included responses and additional results below.**
>
> **NOTE.** We kindly request you to watch this updated video (https://youtu.be/jtFHqFOaQXQ) with more comparisons to baselines, clearly highlighting the advantage of our approach.
>
> **Comparison to graph-based methods.** Thank you for the suggestion. We're happy to include a discussion of works on motion graphs and human motion capture[2-5]. Please let us know if you had something else in mind.
>
> **Comparison to video generation methods.** We compare our method to MoCoGAN [1], an unconditional video generation method. Results are shown on this website (https://sites.google.com/view/iclr2021contrastiveresults). A 3-second video of a candle flame is given as input and as seen, our method is able to produce a 30-second long high quality, dynamically consistent, and diverse output of a candle flickering. On the other hand, MoCoGAN's video contains artifacts, is not high resolution and flame lasts for only about 3 seconds. Similarly, with the guitar video, our method produces a smooth video with seamless transitions whereas MoCoGAN's output is blurry.
>
> [1] "MoCoGAN: Decomposing Motion and Content for Video Generation", Tulyakov et al., CVPR 2018.
>
> [2] "Motion graphs." Kovar et al., ACM SIGGRAPH 2008.
>
> [3] "Motion synthesis from annotations." Arikan et al.,  ACM SIGGRAPH 2003.
>
> [4] "Interactive motion generation from examples." Arikan et al., ACM TOG 2002.
>
> [5] "Interactive control of avatars animated with human motion data." Lee et al., Computer graphics and interactive techniques 2002.

---

### Official Review · AnonReviewer3 · 2020-10-30
**This paper presents Contrastive Video Textures for video synthesis by resampling frames from an input video. The model is video-specific and represents a video as a graph where nodes indicate individual frames and edges are transition probabilities. An output video is generated by traversing edges with high transition probabilities. A deep video interpolation model is used to make visual smoothness for large transitions. This model can be extended to an audio conditioned video synthesis task.**

**Rating:** 4
**Confidence:** 5

**Review:**

** Strengths

(1) Improve the classic video texture synthesis method Video Textures by replacing pixel similarity with a distance metric learning to measure the transition probabilities

(2) Extend the proposed approach to audio conditioned video synthesis

(3) Outperform the competing algorithms on a set of evaluations


** Weaknesses

(1) It seems a strong limitation that the proposed approach is not able to generalize to different videos or has to be video-specific (i.e., train a model on each input video).

(2) It is not true that existing methods fail to generate more than a short sequence of frames, e.g., (Lee et al. 2019) in theory can generate videos with arbitrary lengths.

(3) The pre-trained interpolation network (Jiang et al. 2018) seems a quite important component for the proposed algorithm to generate smooth videos. No analysis is provided to show the contribution of this module.

(4) Lacks of analysis or comparisons to justify some important hyper-parameters, e.g., (1) how the softmax temperature term impacts the synthesis results, (2) how t% is chosen to threshold the transition probabilities.

(5) For video encoding, two separate subnetworks are used to break the symmetry between query and target embeddings, which makes sense considering temporal ordering. However, why audio embeddings take a same encoding network?

(6) For unconditional video synthesis, the musical instrument playing videos mostly contain small motions. Unclear how this approach could handle general videos exhibiting with large actions.

(7) It is claimed that the approach is able to produce infinite video, however the content is constrained in the input video, so the variation is limited.

(8) What are the results if the classic methods also use the interpolation network?

---

> ### Author Response · Authors · 2020-11-17
> **Thank you for the constructive feedback. We've included responses and additional results below.**
>
> **NOTE.** We kindly request you to watch this updated video (https://youtu.be/jtFHqFOaQXQ) with more comparisons to baselines, clearly highlighting the advantage of our approach.
>
> **Generalization.** Our work is similar to test-time training methods such as SinGAN[1], Deep Image Prior[2] which train example-specific models. Our method only takes a few hours to train on a single video and doesn't require hours of training on a large dataset. Further, we show that it works on a diverse set of videos such as dance, musical instruments, and natural scenes.
>
> **Comparison to video generation methods.** We agree that some video generation methods are capable of generating long sequences and we will make the correction. We compare our method to MoCoGAN [3], an unconditional video generation method. Results are shown on this website (https://sites.google.com/view/iclr2021contrastiveresults). A 3-second video of a candle flame is given as input and as seen, our method is able to produce a 30-second long high quality, dynamically consistent, and diverse output of a candle flickering. On the other hand, MoCoGAN's video contains artifacts, is not high resolution and flame lasts for only about 3 seconds. Similarly, with the guitar video, our method produces a smooth video with seamless transitions whereas MoCoGAN's output is blurry.
>
> **Interpolation module.** We'd like to point out that the interpolation module only helps smoothen any jarring transitions that may arise from large changes. Here (https://sites.google.com/view/iclr2021contrastiveresults/home?authuser=2#h.yt2qctwyutke) we compare two clips before and after interpolation. Frames are added by the interpolation module when there's a cut/transition in the synthesized video. As seen, the interpolation smoothens the transitions. We will include a perceptual study comparing results with and without interpolation.
>
> **Hyperparameters.** Appendix Fig 5b shows the variation in the number of transitions with the temperature parameter. In general, a higher temperature and threshold yields more variations in the output. We typically set the temperature to be 0.3, and the threshold to be 0.02. These were chosen heuristically by visualizing results on the train set, as described in Sec 4.
>
> **Audio encoding subnetworks.** We'd like to point out that we only need to break the symmetry when both the query and target segments are from the same signal. The audio subnetwork is used to compare audio segments in the source audio with the audio segment from the conditioning audio. Given a new conditioning audio segment, we find the best matching segment in the source audio by embedding both segments and computing the similarity between them. Since the segments are from different audio signals, there's no need to have two separate subnetworks as there's no symmetry.
>
> **Large motion changes.** We show results on a dance video with large motion changes here (https://sites.google.com/view/iclr2021contrastiveresults/home?authuser=2#h.yluvygyt7gkw). The input is a ~20 min long dance video. We segment the dancer and remove the background in the input video to ensure that the resulting texture is unaffected by changes in background and lighting conditions.
>
> **Limited variation.** We agree that resampling frames from the original video limits the diversity of the output video. However, the constant transitions ensure that the original video is never replayed as is. We measure the diversity as the number of transitions in every 30 seconds of the synthesized video and our method achieves a diversity score of 15.78.
>
> **Classic method with interpolation.** We added interpolation to the classic approach just as we did for the contrastive method. Results can be viewed here (https://sites.google.com/view/iclr2021contrastiveresults/home?authuser=2#h.es1qvmwuhe29). There is no improvement in the quality of the output synthesized by classic even when interpolation is added. Contrastive results look significantly better.
>
> [1] "Singan: Learning a generative model from a single natural image." Shaham et al., ICCV 2019.
>
> [2] "Deep image prior.", Ulyanov et al., CVPR 2018.
>
> [3] "MoCoGAN: Decomposing Motion and Content for Video Generation", Tulyakov et al., CVPR 2018.

---

### Official Review · AnonReviewer2 · 2020-11-01
**A simple method with non-parametric method for long-range video generation, expecting for more comprehensive experimental results and better results.**

**Rating:** 5
**Confidence:** 5

**Review:**

Summary of this paper: In this work, the authors propose a method to learn to generate long-range video sequences. The general idea is starting from a prior work (Video Textures) and extending this work with a learning framework. Specifically, during training a model is used to learn the transition probability between different video segments. During inference, long-range video synthesis is achieved through iterative sampling of new video segments. To guarantee the smoothness of the transition between different segments, an existing interpolation method is used to connect these video segments in a sequential order.

Pros:

+ quality: This paper is overall easy to read. The motivation behind this work is clearly presented, i.e., to synthesize long-range video sequences. In the introduction part, the authors present the basis of their work (Video Textures) and present a comprehensive comparison with previous works. The authors also present sufficient qualitative results to demonstrate the superiority of their work.

+ clarity: The pipeline of the proposed method is clearly presented in the method part. The general framework is very straightforward. The analysis of the quantitative and qualitative results is convincing and logical.

Cons:

- originality: This work is more like a simple extension of the previous work (Video Textures) with limited novelty. I am confused about the difference of the basic formulation of the core  transition probabilities (Eq. 1 and 2). Are they different from the work (Video Textures) or not? It is highly recommended that the authors could present more comparison with the previous baselines in both general idea and model details. Moreover, the video interpolation is directly borrowed from previous work without further improvements, where I think is still challenging and worth to explore. The audio conditioned video synthesis part also seems like an extra module which does not influence the completeness of the whole model if not included.

- significance: I have carefully checked the quality of the generated video sequences, which are not so satisfying. First, these are noticeable discontinuity between sampled video segments. Second, this method seems to be example-specific, which needs retraining if fed a new video sequence. The scalability of this method is limited. Third, the video content is directly sampled from seen sequence, where the diversity is constrained to the given video.

---

> ### Author Response · Authors · 2020-11-13
> **Thank you for the constructive feedback. We've included responses and additional results below.**
>
> **NOTE:** We kindly request you to watch this updated video (https://youtu.be/jtFHqFOaQXQ) with more comparisons to baselines, clearly highlighting the advantage of our approach.
>
> **Limited Novelty.** Our method is significantly different from Video Textures, and both Eq.1 and Eq.2 are introduced by us and not a part of the original formulation. Most importantly, we "learn" the transition probabilities while Video Textures doesn't. We include a detailed explanation of Video Textures in Appendix Sec A.2. Video Textures computes L2 distances on raw pixel values of the frames and uses this to construct a matrix of transition probabilities between frames. In contrast, we extract overlapping segments from the video and use a 3D CNN to learn feature representations for these segments. The network is trained using NCE loss on the task of selecting the correct positive segment given the query segment and target segments. We then use the learned feature representations and similarity metric to compute transition probabilities as described in Sec 2. Further, we'd like to point out Appendix Fig 5a which shows the difference in transition probabilities learned by both methods, clearly highlighting that our method is better at identifying subtle differences between segments.
>
> **Purpose of audio conditioned video synthesis.** The purpose of this extra module is to show that our method can be extended to synthesize videos that match a conditioning audio signal, while Video Textures fails when conditioned on an audio signal. We compare to existing audio conditioned video synthesis works and show that MTurk evaluators constantly preferred the videos synthesized by our method.
>
> **Quality of results and interpolation.** We agree that the results do have obvious cuts in some instances, but the interpolation helps smoothen transitions and the synthesized videos are much better than the classic method, as seen [here](https://sites.google.com/view/contrastivevticlr2021/#h.pnlaoxsj4yau). Using a GAN based method for interpolation is an interesting direction for future work.
>
> **Generalization.** Our work is similar to test-time training methods such as SinGAN[2] and Deep Image Prior[3] which train example-specific models. Our method only takes a few hours to train on a single video and doesn't require days of training on a large dataset. Further, we show that it works on a diverse set of videos such as dance, musical instruments, and natural scenes.
>
> **Diversity.** We agree that resampling frames from the original video limits the diversity of the output video. However, the constant transitions ensure that the original video is never replayed as is. We measure the diversity as the number of transitions in every 30 seconds of the synthesized video and our method achieves a diversity score of 15.78.
>
> **Additional results.** We present additional results here (https://sites.google.com/view/iclr2021contrastiveresults). We compare our method to MoCoGAN [1], an unconditional video generation method. A 3-second video of a candle flame is given as input and as seen, our method is able to produce a 30-second long high quality, dynamically consistent, and diverse output of a candle flickering. On the other hand, MoCoGAN's video contains artifacts, is not high resolution and flame lasts for only about 3 seconds. Similarly, with the guitar video, our method produces a smooth video with seamless transitions whereas MoCoGAN's output is blurry. We also show the advantage of using interpolation and include more comparisons to the classic Video Textures work.
>
> [1] "MoCoGAN: Decomposing Motion and Content for Video Generation", Tulyakov et al., CVPR 2018.
>
> [2] "Singan: Learning a generative model from a single natural image.", Shaham et al., ICCV 2019.
>
> [3] "Deep image prior.", Ulyanov et al., CVPR 2018.

---

### Decision · Program_Chairs · 2021-01-07
**Final Decision**

**Decision:**

Reject

**Comment:**

The paper initially had mixed reviews (4,5,6).  The main issues raised were:
1) limited novelty (re-using/integrating components) [R2];
2) limited generalization ability since the model needs to be retrained on every video [R2, R3];
3) limited applicability - experiments limited to certain domain of video, while results on videos with large motion are not convincing [R2, R3];
4) missing ablation studies / experiments [R3, R4].

The author response partially addressed some concerns, but the main points 1-3 are still problematic. In addition, the AC noted that the technical aspect was lacking:

- Training with contrastive loss on a single video may likely overfit the embedding to the video, which leads to a meaningless embedding where all non-neighboring segments are orthogonal in the embedding space. While changing the softmax temperature can yield higher entropy transition probabilities, the induced probability distribution is probably highly noisy. It would be better to train this on a large video corpus, which will prevent overfitting. Also contrastive loss is typically used to build a discriminative embedding space for classification/recognition, not a smooth embedding space for generation (where distances between embedding vectors are strongly correlated to similarity). Thus some other embedding smoothness terms could be added during contrastive learning.
- The learning is only on the transition probabilities, while the video generation is separate. It would have been more convincing to learn the transition probabilities with the video generation process in an end-to-end manner. Perhaps a discriminator could be placed after the video generator so that the transition probabilities could be learned so as to better mimic real video. Other loss terms based on video temporal smoothness could also be added ensure smoother transitions between clips (e.g., motion consistency).

The negative reviewers remained unconvinced by the author response, and the AC agreed with their concerns. Thus, the paper was recommended for rejection.